# A Manifold Learning Perspective on Representation Learning: Learning Decoder and Representations without an Encoder

**DOI:** 10.3390/e23111403

**Published:** 2021-10-25

**Authors:** Viktoria Schuster, Anders Krogh

**Affiliations:** 1Center for Health Data Science, University of Copenhagen, 2200 Copenhagen, Denmark; viktoria.schuster@sund.ku.dk; 2Department of Computer Science, University of Copenhagen, 2100 Copenhagen, Denmark

**Keywords:** representation learning, manifold learning, neural networks, autoencoders

## Abstract

Autoencoders are commonly used in representation learning. They consist of an encoder and a decoder, which provide a straightforward method to map *n*-dimensional data in input space to a lower *m*-dimensional representation space and back. The decoder itself defines an *m*-dimensional manifold in input space. Inspired by manifold learning, we showed that the decoder can be trained on its own by learning the representations of the training samples along with the decoder weights using gradient descent. A sum-of-squares loss then corresponds to optimizing the manifold to have the smallest Euclidean distance to the training samples, and similarly for other loss functions. We derived expressions for the number of samples needed to specify the encoder and decoder and showed that the decoder generally requires much fewer training samples to be well-specified compared to the encoder. We discuss the training of autoencoders in this perspective and relate it to previous work in the field that uses noisy training examples and other types of regularization. On the natural image data sets MNIST and CIFAR10, we demonstrated that the decoder is much better suited to learn a low-dimensional representation, especially when trained on small data sets. Using simulated gene regulatory data, we further showed that the decoder alone leads to better generalization and meaningful representations. Our approach of training the decoder alone facilitates representation learning even on small data sets and can lead to improved training of autoencoders. We hope that the simple analyses presented will also contribute to an improved conceptual understanding of representation learning.

## 1. Introduction

The original article on backpropagation is called “Learning Internal Representations by Error Propagation” [1], and indeed, learning in neural networks can be viewed as the learning of intermediate representations in the different layers. The neural network thus performs a transformation of the input through a series of these internal representations. In representation learning (reviewed in [2,3]) the objective is to use these learned representations for other applications as they can for instance map discrete high-dimensional input samples like text to a Euclidian space of lower dimensionality and hopefully learn (or preserve) relatedness by assigning similar representations to related examples. Feed-forward autoencoders, whose objective is to reproduce the input on the output layer, are often used for unsupervised (or “self-supervised”) representation learning, although one can of course learn representations even when inputs and targets differ.

In the related field of manifold learning, the objective is likewise to find a representation of samples. It is assumed that the high-dimensional data lie on a lower-dimensional manifold, and the aim is to construct a map of the (training) data on such a manifold. Principal component analysis (PCA) is the simplest and most used method, in which the linear subspace explaining most of the variation in the data is found. The main difference between manifold learning and representation learning is that in the latter one obtains an encoder that maps from the input space to the representation space and a decoder that maps from the representation space back to the input space. Additionally, the objective function in manifold learning often takes neighbourhood relations into account, whereas samples in standard neural network training are treated independently. The relationship between representation learning and manifold learning has been extensively discussed in works such as [2,4,5].

Autoencoders were originally introduced in [1], with image compression [6] and speech recognition [7] as some of the first applications. In [8] it was shown that simple auto-encoding feed-forward neural networks, with a single hidden layer and optimized to reproduce real-valued input on the output, will converge to the principle components subspace in the hidden layer. Since then, there has been significant progress with respect to regularization and robustness of autoencoders, such as using noisy inputs to train denoising autoencoders [4] and making the contractive autoencoders aiming at more robust encoding [9]. The relation between autoencoders and manifold learning has been discussed before, see e.g., [4,5].

Here, we took up a simple but not too well-known view on autoencoders and representation learning from a manifold learning perspective. We showed that the decoder maps from the representation space to a manifold in the input space. We then showed how a decoder can be trained without an encoder by optimizing the representations of the training data directly together with the weights of the decoder similar to manifold learning. In training, we optimized towards a manifold that has the minimum distance between the training points and their projections onto the manifold. Learning of a representation along with an autoencoder has been introduced before as predictive sparse decompositions [10], and others have even made the step of separating representation and the decoder from the encoder [11]. We wished to take up this view on representation learning and show its benefits in terms of simplicity, performance, and data-efficiency.

We derived expressions for the number of samples needed to specify an encoder and a decoder and showed that in most situations, the decoder is much better specified than the encoder. The theoretical predictions were confirmed on three different data sets. We demonstrated how training of the decoder alone performs much better than a standard autoencoder on small data set sizes, while performance on larger data sets is still better but very close to the autoencoder. We further showed that the learned representation can be useful for downstream tasks and that specific solutions can potentially be derived better from the decoder than the autoencoder.

This study was basic and straightforward mathematically, and it is our hope that it will help readers build intuition about decoders, encoders, and autoencoders in this manifold learning perspective.

## 2. Encoder-Free Representation Learning in Theory

### 2.1. The Linear Case

Assume the data are *n*-dimensional real vectors. In PCA, a linear subspace is obtained in which the first basis vector points in the direction of highest variance, the second points in the direction of highest variance of the subspace orthogonal to the first, and so on. When PCA is used for manifold learning and dimensionality reduction, the first m<n principal components are used, and the data points are projected onto this linear subspace of dimension *m*, which we call the principal subspace.

If the basis vectors of the principal subspace are called ${\vec{w}}_{i}$, the projection of a point $\vec{x}$ can be written as $\sum_{i=1}^{m}z_{i}{\vec{w}}_{i}$, where $\vec{z}$ is the *m*-dimensional representation of $\vec{x}$ and $z_{i}=\vec{x}\cdot {\vec{w}}_{i}$. The *z* vectors are analogues of the representations in representation learning or manifold learning.

If we assume that data have mean zero, the linear subspace is also the one which has the smallest mean distance to the data. Therefore, the principal subspace can be found by minimizing the mean distance between the data points and their projections,
(1)$$\sum_{k}\left|\right|{\vec{x}}^{k}-\sum_{i=1}^{m}z_{i}^{k}{\vec{w}}_{i}{\left|\right|}^{2},$$
where *k* indexes the data points. (We do not give the detailed proof, but it is relatively straight-forward to expand the square in (Equation 1) and using $z_{i}^{k}={\vec{x}}^{k}\cdot {\vec{w}}_{i}^{k}$ to show that minimizing (Equation 1) corresponds to maximizing the variance $\sum_{i,k}{\left({\vec{x}}^{k}\cdot {\vec{w}}_{i}\right)}^{2}$ when requiring that the ${\vec{w}}_{i}$s are orthonormal.) This will not normally give an orthonormal basis, but vectors *w* will still span the principal subspace. We can recognize this as a linear “neural network” with weights *w*, corresponding to the decoder part of a linear autoencoder, mapping from a representation *z* to a point *x* in input space. It is a classic result that a linear autoencoder will learn the principal subspace [8].

Note that (Equation 1) does not have a unique solution. The weights and representations can be scaled arbitrarily (wx=(w/s)(zs) for a constant *s*), and it is invariant to permutations of the order of vectors and rotations within the subspace in general. One could apply normalization or impose other constraints on the solution to limit the freedom.

### 2.2. Non-Linear Decoders

Assume now that we have a non-linear mapping that maps a representation *z* to a point $g_{w}\left(z\right)$ in input space (we dropped the vector arrows for ease of notation). We assume also that $g_{w}$ is a continuous function that depends on some parameters *w* and possibly some form of regularization. We assumed m<n, $g_{w}$ defines a manifold in input space of dimension *m* (or lower), onto which all points in the representation space are mapped.

Above, we saw how one can obtain the principal subspace by minimizing the distance between the data points and their projections. We can do the same for a non-linear decoder. The idea is then to find the manifold defined by $g_{w}$ that minimizes the mean distance (or the loss) between the training points and their projections onto the manifold, $L(x,g_{w}\left(z\right))$, where *L* is the loss function and *z* and *w* are parameters. See Figure 1 for an illustration. (Here “projection” means the point on the manifold that we map a point to, so it is not used in a strict mathematical sense).

Usually, internal representations are learned implicitly by adapting the weights of the neural network, but there is nothing stopping us from treating them as parameters and learning them explicitly together with the weights (see e.g., [12]). A loss function $L(x,\hat{x})$ used to train an autoencoder, where $\hat{x}$ is the autoencoder output, can be used with internal representations. The loss for input *x* would be $L(x,g_{w}\left(z\right))$, where the representation *z* is unknown and found together with the weights by minimizing the total loss. So, for each of *N* training examples, ($x^{1}\ldots x^{N}$ ), the representation is optimized during training, so the total loss
(2)$$\sum_{k}L\left(x^{k},g_{w}\left(z^{k}\right)\right)$$
would be minimized with respect to both *w* and the vectors $z^{1}\ldots z^{N}$. This can be done by any optimization technique, such as gradient descent. Gradients with respect to *z* are simply the back-propagated errors that we are calculating anyways for weight optimization. As in the linear case described above, there are scaling and symmetry invariances, so the solution is not unique. It is therefore advisable to constrain the weights (for example, via a small weight decay).

If the decoder is a single layer, there is only one optimal solution (apart from scaling and symmetry operations), and the optimization problem is convex, i.e., it is very easy to learn, and it would essentially recover the PCA if the layer is fully connected. With a multi-layer decoder, one could learn the internal representations layer-by-layer using convex optimization, but this is unlikely to lead to an overall optimal solution and was not considered further here.

Once the weights of the decoder and training set representations are determined by training, how can we use it for new samples, e.g., in a test set? One possibility is to do the same optimization by gradient descent with fixed weights to find the optimal representation for the new sample. This was our approach when reporting results of the decoder alone. A disadvantage of this approach is that there may be local minima trapping the optimization, and thus it may not give a globally optimal representation. In our experiments, we did not observe problems relating to this. An alternative is to train an encoder, as we discuss later.

One disadvantage of optimizing the representations directly within the decoder training loop is that each sample only receives one gradient update per epoch, whereas the weight updates are based on gradients from the whole data set. One could therefore suspect a large number of epochs is needed for the representations, but this was not a major issue in our experiments. If the training set is very large, one could train the decoder on a sub-sample and afterwards train an autoencoder on the whole dataset (see below).

### 2.3. The Number of Samples Needed to Train the Decoder

In an idealized noise-free case, the decoder should fulfil $x=g_{w}\left(z\right)$ for all *N* training examples. If *g* is a linear function, there exists an exact solution for *w* and *z* if the number of weights $C_{d}$ plus the number of parameters for the representation, Nm, is larger than the number of constraints Nn: one equation per example (*k*) per output unit (*i*), $x_{i}^{k}=g_{w}\left(z^{k}\right)$. We defined the load as the number of constraints per parameter,
(3)$$\alpha_{d}=\frac{Nn}{C_{d}+Nm}=\frac{n}{m}{\left(1+\frac{C_{d}}{Nm}\right)}^{-1}.$$

If the load is below 1, the system is under-determined, and if it is above 1, it is over-determined (if the training vectors are linearly independent).

For non-linear decoders, such as multi-layer perceptrons, we expect the above to be approximately true, although $C_{d}$ should be replaced by an “effective” number of parameters, which we call the network complexity. Formally, we defined this complexity as the number of training examples needed to train a unique model apart from scaling and symmetry operations. There is currently no theory to calculate this complexity, and we approximated it by the number of weights. The number of weights is an upper bound for the effective number of parameters, and therefore, the load, as defined above, will generally be underestimated, and we are thus less likely to apply too large a model to the given data.

In most realistic situations, we would want the system to be over-determined, i.e., $\alpha_{d}>1$, because otherwise the representations and the decoder would adapt to the noise (alternatively, the system can be regularized). When the number of weights is less than Nm, the load is essentially the input dimension divided by the representation dimension, the compression rate n/m, which is normally much larger than 1 when the objective is to find a low-dimensional representation of high-dimensional data. When n>>m, it is generally quite easy to obtain high loads. (For m>n, the load is always below 1 and is only relevant with sparsity constraints or other regularization.) Another way to interpret this is that you would be wise to either construct your neural network decoder to have a relatively large $\alpha_{d}$ or use some form of regularization.

We have seen that one can learn a manifold in input space using a neural network decoder, and it is therefore a form of manifold learning. The mathematical properties of the manifold are determined entirely by the neural network decoder, and the learned manifold may be further constrained by regularization. If the decoder is over-determined (α>1), the training data cannot normally be contained in the manifold exactly, and minimization of the loss will find the manifold that is closest to the training data (closest in the sense of having minimum loss).

### 2.4. The Encoder and Autoencoder

We see that in theory the encoder is not really needed for learning low-dimensional representations. However, finding the representation of a new data point *x* requires a minimization of $L(x,g_{w}\left(z\right))$ in *z* with fixed weights in the decoder. Although this can easily be done, it is often desired to have an encoder instead that maps directly from input space to representation space. In an autoencoder, the encoder puts additional constraints on the representations, meaning that it will not necessarily learn the same representations as the decoder would recover on its own.

In the framework of manifold learning, the encoder can only make things worse, because with the chosen parametrization of our manifold (the decoder) and the chosen distance metric (the loss function), the representations found using only the decoder are optimal—they minimize the loss and the distance to the manifold. In representation learning, one could instead view the encoder as a possibility to impose constraints on the representations as a sort of regularization. However, regardless of the encoder, reconstructions will lie on a manifold defined by the decoder.

In principle, the encoder could be of such high complexity that it could learn almost exactly the “true” decoder representations. If it had high enough complexity, we know that the “true” representations would be found, because they minimize the loss. However, when the encoder complexity is very high, it is again likely that we will not have enough data. Instead, once the representations are estimated (along with the decoder), the encoder can principally be trained using the learned representations as targets. For an encoder, the load is therefore
(4)$$\alpha_{e}=\frac{mN}{C_{e}}$$
when the representations of the training examples are fixed (Nm equations/constraints on the weights). Interestingly, there is no simple relation between this load and the decoder load. If the encoder and decoder have the same complexity $C_{e}=C_{d}$ and m<n, we will have from (Equation 3) and (Equation 4) that
(5)$$\alpha_{d}=\frac{n}{m}{\left(1+\alpha_{e}^{-1}\right)}^{-1}=\frac{n}{m}\frac{\alpha_{e}}{1+\alpha_{e}}.$$

For small $\alpha_{e}$, $\alpha_{d}$ is n/m times larger, and when $\alpha_{e}$ is large, $\alpha_{d}$ is essentially equal to n/m. Often, the compression factor n/m is in the hundreds, and the decoder would thus be “hundreds of times more well-specified” than the encoder; in many situations, one would have plenty of examples to learn the decoder but not enough for the encoder.

The above analysis is a little simplified. Since the assumption is that the training data lie close to an *m*-dimensional manifold in input space, the data were not scattered randomly, and it is likely that the encoder can be trained on a smaller number of samples in this region of sample space. In other words, it may be possible to train the encoder with a small load for points in or close to the manifold. However, for outliers, an encoder trained with a small load is likely to give arbitrary results and thus the autoencoder reconstruction will be far from the projection on the manifold. We thus expect that out-of-distribution samples will be poorly reconstructed by an autoencoder compared to a decoder.

Several methods have been proposed that increase the robustness of autoencoders regarding over-fitting. One such method is the denoising autoencoder [4] in which noise is added to the input samples and in which the autoencoder is trained to reconstruct the noiseless version. In the present perspective, this seems like an excellent approach, because this will minimize the distance between noisy points and their projections. The contractive autoencoder [9] imposes regularization on the encoder that favors similar inputs to have similar representations. This is done by adding a regularizer term with the squared derivatives of the encoder with respect to the *x* (the Frobenius norm of the Jacobian). Zero gradients imply orthogonality with the manifold, and this approach therefore favors an orthogonal projection of points onto the manifold.

Both denoising and contractive autoencoders have the desired effect of making the encoder more robust, but they have the less desirable side-effect of also trying to minimize variation within the manifold. Therefore, we would suggest a variant of the denoising autoencoder in which the decoder is trained as above and the encoder is trained on noisy examples with its “true” representation as the target, that is, the representation defined by the decoder. This is essentially equivalent to training the autoencoder with *fixed decoder* and using the noisy examples both as input and output. This approach will train the decoder to give the correct projection of the out-of-distribution input onto the manifold learned by the decoder on the training set. It should thus be possible to use much higher noise levels than in denoising autoencoders, and in principle, one can train on completely random data, once the decoder is fixed. Care must be taken, however, to ensure that the encoder still encodes the training data well.

## 3. Hypothesis Testing on Real and Simulated Data

In this section, we aimed to answer the following questions: can a decoder on its own generalize well on small data set sizes, whereas an autoencoder overfits as hypothesized? Is the resulting representation meaningful/useful for downstream tasks? Does the decoder work on larger data sets?

We demonstrated that a representation can be learned by a decoder alone using three different data sets. The first two are popularly used image data sets MNIST [13] and CIFAR10 [14], and the last one is a simulated data set as an example of regulatory data with a known and desired representation.

All networks were implemented in Python using PyTorch [15], and Jupyter notebooks are available with the code. Most runs were done on Google Colaboratory. Details about model architectures for all experiments covered in the following sections can be found in the Appendix B.

### 3.1. MNIST

In our first experiment, we used the standard MNIST data set [13] and trained on random subsets of varying sizes from 500 to 15,000 examples using a decoder on its own and a naive autoencoder. The encoder of the autoencoder consists of two convolutional layers followed by a fully connected representation layer. The convolution layers both have 2D kernels of size 4, stride 2, and 64 channels. The decoder is the reverse with the representation layer fully connected to two layers of transpose convolutions. The representation layer has size 20 with linear output. The output layer uses a sigmoid transfer function, and the other layers use a rectified linear unit (ReLU) [16,17] activation. The loss function is the binary cross entropy and network weights were trained with the Adam optimizer [18] using a learning rate of 0.001 and a weight decay of $1.e^{-5}$. When training without the encoder, the representations were optimized with a stochastic gradient descent [19] with a learning rate of 0.02 and a momentum of 0.9. All networks were trained for 200 epochs. Networks were tested on the same random subset of 1000 examples from the MNIST test set. For the decoder alone, representations of test data were found using the same gradient descent as in training (but with fixed decoder weights, of course).

In Figure 2, we show the training and test error for a decoder and an autoencoder trained on subsamples of the MNIST dataset. We could see that the training errors were very similar for the two models, although smallest for the decoder, as would be expected, because the autoencoder is further constrained by the encoder. For small data set sizes, both models over-fit, but the autoencoder was significantly worse than the decoder alone, which supports our hypothesis. From the other graph, we could observe that the decoder load was above 1 for almost all data set sizes, and from a load around 9 (4000 examples), the test error was stable. The load of the decoder was below 1 for all sizes except the last (15,000) and we see that the test error is still decreasing.

This example shows that although the training errors are comparable, the full autoencoder over-fits the data to a larger extend than the decoder, especially for small sample sizes. The decoder test error was almost constant from around 4000 samples.

In this experiment, we also trained an autoencoder with the trained decoder fixed. This performed much worse than the decoder and the autoencoder (results available in the Appendix A). Although unexpected at first, we interpreted it in the following way: the full autoencoder can find a good solution that satisfies the constraints of both the decoder and the encoder, reaching a compromise between them. When the decoder is fixed, however, the encoder is forced to learn the representations dictated by the decoder. For small data sets, the encoder can learn with a small training error but has a high test error. For larger sets, when the load of the encoder approaches 1, both the training and test error increase due to the constraints imposed by the decoder.

### 3.2. CIFAR10

After initial promising results from the simple task of learning a representation for MNIST, we aimed to more systematically demonstrate the efficiency and validity of our proposed approach using the more complex CIFAR10 data set [14]. Our aim was to compare the performance of a simple decoder to an autoencoder with an identical decoder and a symmetric encoder on natural image data. We trained both the single decoder and the autoencoder and compared the reconstruction capabilities of these models for different size training set sizes.

The architectures of our models are based on work from [20]. In order to find a good decoder architecture, we ran a model search whose search space we restricted based on prior knowledge derived from [20,21]. In [21], a thorough investigation of convolutional architectures based on residual bottleneck blocks introduced by [22] was conducted, resulting in guiding principles for a limited design space of convolutional networks for image classification. The initial architecture taken from DCGAN [20] represents the generator. A Appendix A provides a report of our model optimization and a description of how we arrived at using an altered version of the DCGAN generator as our decoder.

The decoder consisted of five 2D transposed convolutional layers with kernel size 4, stride 1, and an output layer with kernel size 1 and stride 1. The first and last transposed convolutional layers had padding 0, while all others had padding 1. For constructing channel sizes, we used a basis of 64, which we called the capacity. The representation (the input of the decoder) was a 1D vector of length 256, which was four times the capacity. This was reshaped as the decoder input to a tensor of 256×1×1. The channel sizes were reduced to capacity×2 (128) in the first layer, capacity (64) in the third, and 3 (output channel size) in the last layer. Batch normalization was applied between all layers, as well as ReLU activation. The last convolutional layer followed a sigmoid activation due to the normalization of the data. The corresponding autoencoder consisted of the decoder architecture and a mirrored encoder with convolutional layers instead of transposed convolutional layers. We refer to this decoder and autoencoder as decoder_1×1 and autoencoder_1×1 (or AE_1×1 in images due to limited space), respectively, to highlight the decoder’s input pixel dimension.

In order to demonstrate that a decoder on its own outperforms a comparable autoencoder on small data sets, we trained decoder_1×1 and autoencoder_1×1 on three class-balanced subsets as well as the full training set (train–test split provided by the data) five times with different random seeds (see Appendix A). The resulting numbers of train images per class were 50, 100, 500, and 5000. Other training parameters included the Adam [18] optimization with weight decay $1e^{-5}$, the learning rate $1e^{-4}$, the representation learning rate $1e^{-1}$ (representations were optimized with stochastic gradient descent [19]), the mean squared error (MSE) loss, and the training time of 100 epochs. The test error was reported on a tenth of the test set (class-balanced, same set for all runs).

In Figure 3A,C we can see that the train sample size had a much lower effect on the test reconstruction error in decoder_1×1 compared to autoencoder_1×1. While both models’ errors converged for the full training set size, autoencoder_1×1’s test loss was approximately three times higher than that of the decoder for the smallest train sample size of 50 images per class. Figure Figure 3A shows that this difference in test loss was not a result of insufficient training time. More precisely, this figure further supports our hypothesis of the relationship between decoder and encoder load. As shown in the Appendix A, the decoder load was above 1 for all training subsets, while the encoder load achieved values above 1 only for the full train sets. Since mere reconstruction loss alone is not a satisfactory evaluation of a model’s performance, we also trained decoder_1×1 and autoencoder_1×1 for 400 epochs each on the full train set and evaluated them on the full test set with random seed 0, saving model parameters every 50 epochs. Figure 3E shows reconstructions of the first eight test images (originals shown in Figure 3D) of both models after 50, 200, and 400 epochs. While the decoder_1×1’s MSE loss for the test set was above that of autoencoder_1×1 until roughly epoch 200 and while the reconstructed images are blurry, they were much more recognizable than those of autoencoder_1×1 at any of the selected epochs. These results show that this convolutional decoder is superior to its equivalent autoencoder in reconstructing images from a low-dimensional representation for small data sets and that it can more than keep up with the autoencoder for larger data sets where the encoder load is sufficient.

However, we are aware that designing and optimizing a decoder mapping from a low-dimensional representation and then comparing its performance to its equivalent autoencoder could be biased in favour of the decoder. We thus conducted a second experiment for which we optimized an autoencoder architecture and training parameters completely disregarding the decoder performance in the optimization step. The model development is included in Appendix A. The architecture of the best performing autoencoder was similar to the previous architecture of autoencoder_1×1, but the last encoder and first decoder convolutional layers were removed, resulting in a representation of dimension channel×4×4. We refer to these architectures as autoencoder_4×4 and decoder_4×4. The other change to the previous architecture was the number of channels in the convolutional layers. With a capacity of 16, the decoder (and symmetrically the encoder) took the representation vector of length 1024 as input of dimension 64×4×4. Each layer’s output channels were of size 64, 32, 16, and 3, respectively.

We trained autoencoder_4×4 and decoder_4×4 on the full train set for 400 epochs with (coincidentally) the same hyperparameters as for training 1×1 models. The test learning curves for both models are shown in Figure 3B along with those of the 1×1 models for comparison. The 4×4 learning curves show a similar but elongated behavior to those of the 1×1 models. Test losses for decoder_4×4 and autoencoder_4×4 converged to much lower levels and converged to similar losses at the end of the 400 epoch training period. This indicates an accelerated learning of autoencoder_4×4. We again reconstructed images after 50, 200, and 400 epochs, which are depicted in Figure 3F. These show an extensive improvement in the autoencoder reconstruction capabilities from autoencoder_1×1 to autoencoder_4×4. Images appeared quite sharp but showed inferior color reconstruction compared to images from decoder_4×4.

Even though the convergences for decoders trained on the full data set for 400 epochs compared to the autoencoders were slower, Figure 3D,E shows that the decoders achieved qualitatively better reconstructions, especially for lower-dimensional representations. When comparing a decoder with learned representation and an autoencoder with equivalent architecture, this clearly demonstrates that the decoder alone with learned representation poses a better solution than the autoencoder.

### 3.3. Simulated Data

Our last test used simple simulated, almost linear, data. They were inspired by a simplified model of a biological regulatory network that we studied in another project. The regulatory network is built of proteins *Z* that interact with genes *X* and govern their expression. These proteins are called transcription factors and present one of the main actors in gene regulation. They are sequence-specific DNA-binding proteins, which modulate expression by binding close to specific genes. We assumed for simplicity that the regulation is direct and that transcription factors operate independently. We thus assumed that the expression data are governed by the transcription factor levels, *Z*. Since *Z* are sequence-specific, they do not interact with all genes, and with those they do interact, they do not do so equally. We therefore viewed the regulatory network as a weighted bipartite graph, directed from *Z* to *X*. All this can be expressed in
(6)$$x_{i}=ReLU\left(\sum_{j=0}^{n}a_{i,j}w_{i,j}z_{j}+\epsilon_{i}\right),$$
where $a_{i,j}$ is a sparse connectivity matrix and $w_{i,j}$ the strength (and sign) of regulation. To generate the expression vectors x of length n, regulator level vectors z of length m (*m* < *n*) were initialized from random samples of a gamma distribution. The weighted bipartite graph was expressed by the product of an adjacency matrix A (random binary values assigned based on a defined fraction of connections) and a weight matrix W (randomly sampled from a uniform distribution). Noise was added from a normal distribution with mean 0 and sd 0.2. X was ensured to be *R*${}_{\left(>=0\right)}$ through the rectified linear unit (ReLU) [16,17].

We approximated the linear relationship between the regulatory representation *Z* and expression $R_{\geq 0}$ input space *X* via a single-layer decoder. Since part of the objective was to learn the input space and representation as an assignable bipartite graph, it was necessary to apply constraints enabling the hidden units in the representation to be identified as specific regulatory factors, and we therefore assumed a sparsely connected network defined by the adjacency matrix *A*.

Given this assignability of the representation, we were able to restrict the space of possible valid representations towards one close to the regulatory space used to generate the data. Hence, we enabled ourselves to judge whether a meaningful representation was learned that represents a unique feature of the data. As a measure, we used the Pearson correlation coefficient (PCC) between true regulatory vectors and learned representations. We investigated data reconstruction and representation correlation for different decoder loads on three models. These were the decoder, the encoder trained on the pre-trained decoder, and an autoencoder (AE) consisting of the architectures of the single models.

The decoder and the encoder both consisted of a single linear layer (sparse in case of the decoder as described above) with ReLU and leaky ReLU (slope 0.1) as activation functions, respectively. The autoencoder was comprised of a combined encoder and decoder. Different loads were achieved by varying the number of training samples *N* for a constant input space of dimension n=1000 and hidden dimension m=100. Used values for *N* can be found in Appendix A. The data in this experiment were simulated with zero noise and a connectivity of 0.1 (90% of values in A are zero). Training parameters included a mini-batch size of 32, a weight decay of $1.e^{-5}$, and a set of learning rates, which were obtained from a small-scale grid search experiment included in Appendix A. Decoder and representation received learning rates of 0.001 and 0.01 (with momentum 0.9), respectively. The encoder and autoencoder received a learning rate of 0.0001. Weights were optimized with the Adam optimizer [18] and representations with the stochastic gradient descent [19]. All metrics were reported on the test set of sample size 100. For each new training set, a new test set was created. Models were trained and evaluated on the same data sets. The encoder refers to an autoencoder using the pre-trained decoder whose weights were frozen, and autoencoder refers to an identical autoencoder but without pre-training the decoder. The decoder and the encoder were trained for 500 epochs each, and the autoencoder was trained for 1000 epochs.

Figure 4 shows the test reconstruction loss and the representation correlation of the decoder, the encoder (trained on pre-trained decoder), and the autoencoder (AE) for different decoder loads $\alpha_{d}$. We observed that the load $\alpha_{d}$ must be >1 for the decoder to be consistently well-determined and thus for achieving an exact solution for the representations (a correlation of 1) and good input reconstruction. The encoder with a fixed and pre-trained decoder in this experiment took a slightly higher $\alpha_{d}$ of at least 3.3 in order to achieve correlations ≥ 0.99. This may be explained by the higher necessary load of the encoder, which was mostly mitigated here by pre-training the decoder. One reason for a successful training of an encoder on a pre-trained decoder here, unlike in the MNIST experiment, could be that the encoder in this experiment was more complex than the decoder. The naive (not pre-trained) autoencoder in this experiment was unable to find the exact solution for *z*, as can be seen by the generally lower and more variable representation correlations. Additionally, it took a load of roughly 9 for the autoencoder to achieve similar reconstruction losses as the decoder and the encoder, which here was equivalent to training sample sizes above 10,000. These observations support the hypothesis that the encoder is the limiting factor in learning a precise mapping from the representation *z* to the input space *x* and that the decoder on its own can learn a meaningful representation.

## 4. Conclusions

In this study, we introduced a new manifold learning perspective on autoencoders. We argued that it is useful to view the decoder as defining a low-dimensional manifold in input space. Training of a decoder alone amounts to optimizing the weights of the decoder and the representations of the training data so as to minimize the average distance between the training samples and their projections onto the manifold, which are the decoder reconstructions.

We showed that it is possible (and probably common) to have an over-determined decoder and an under-determined encoder. Our tests confirmed that a decoder trained alone generally performs better on the test data than an autoencoder trained from scratch. However, in one test, we saw that an autoencoder trained with a fixed optimized decoder performs almost as well as the decoder alone. We further demonstrated that this approach can lead to meaningful representations that may be useful for downstream tasks. Albeit only covering the task of reconstruction in this demonstration, we believe that this approach can easily be extended to other tasks in need of a useful representation. One such example is time series forecasting, based on building blocks such as LSTMs and transformers.

The similarity between representation learning, autoencoders, and manifold learning has often been pointed out. In most work on autoencoders, the focus has been on the encoder, and various types of regularization have been proposed to limit over-fitting of data, such as [5,9]. In our view, a better understanding can be obtained when first focusing on the decoder, which constrains the autoencoder, because it is generally much better specified by the training data than the encoder.

In this work, we focused on the reconstruction error as a measure of performance. Often, learned representations are used for classification or other tasks. We will extend these ideas to such other tasks in the future to see if the good performance of decoder training extends to these.

The results and observations presented in this study are additionally of high relevance for generative models. We are currently working on the advancement of the ideas in this context. 

## Figures and Tables

**Figure 1 entropy-23-01403-f001:**
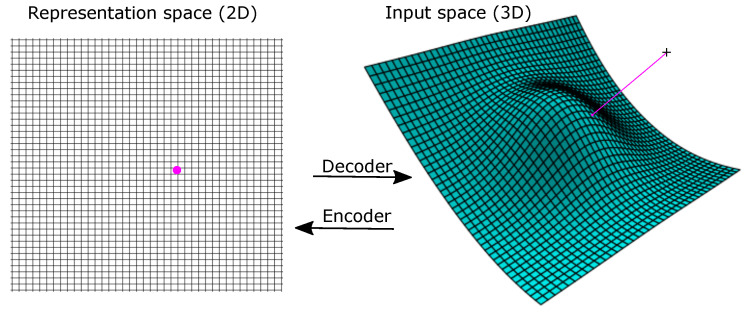
The decoder maps from the low-dimensional representation space to a manifold in input space. Here the representation space is 2D, and the input space is 3D. The representation of a point in input space is the point in representation space that maps to the nearest point on the manifold. Here, it is illustrated in 3D, where the point in representation space to the left maps to the point on the manifold that is closest to the point at the cross.

**Figure 2 entropy-23-01403-f002:**
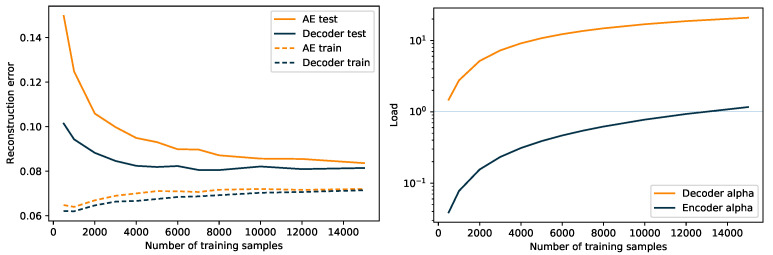
**Left**: Training and test error for a decoder and an autoencoder trained on subsamples of different sizes (x-axis) of the MNIST data. Details of the networks are given in the text. **Right**: The load of the encoder and decoder vs. the training set size.

**Figure 3 entropy-23-01403-f003:**
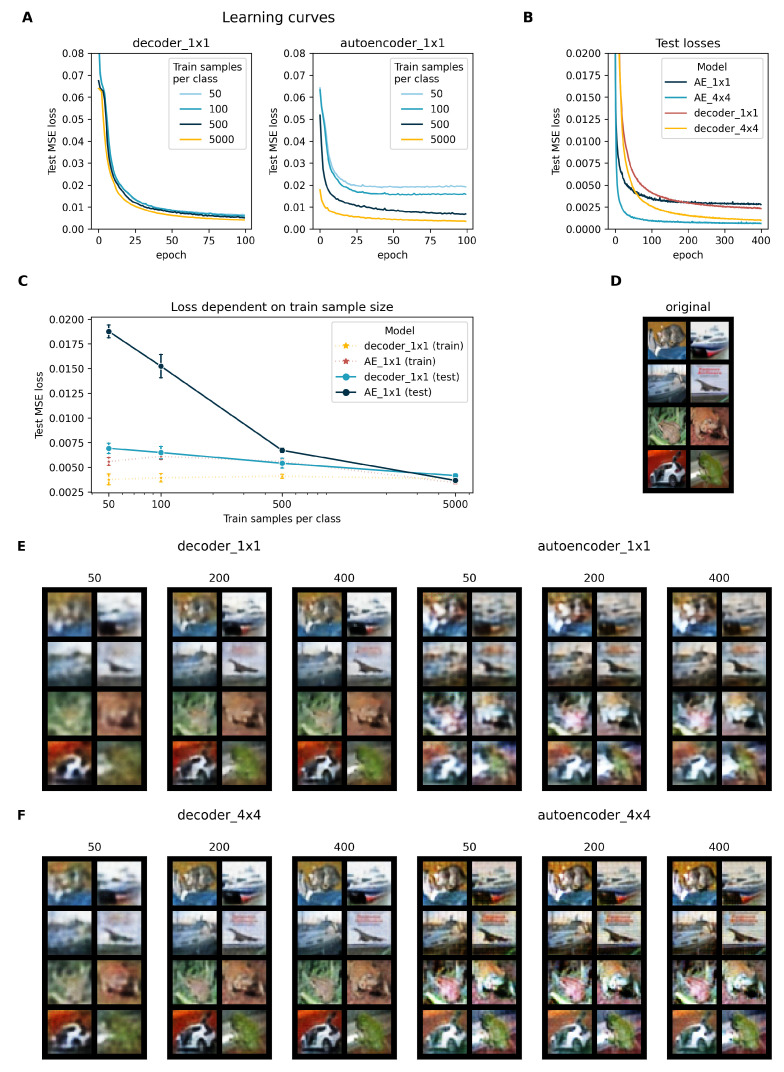
Results on CIFAR10. (**A**) Learning curves of decoder_1×1 and autoencoder_1×1 for different training sample sizes (indicated by color). The y axis reports test reconstruction losses as mean squared errors. All models were trained for 100 epochs and initialized with random seed 0. (**B**) Learning curves of decoder and autoencoder with different representation sizes. (**C**) Reconstruction error for decoder_1×1 and autoencoder_1×1 trained on class-balanced subsets of CIFAR10 plotted against the per-class sample size of the train set. Means and standard deviation error bars were derived from 5-fold replication with different random seeds. Dotted lines with asterisk markers refer to train loss, full lines, and round markers to the loss on the full test set. (**D**) The first 8 images of the CIFAR10 test set. (**E**) Reconstructed test images of models with representation 256×1×1. Numbers above the image grids indicate the training time in epochs. The first three grids show reconstructions from decoder and representation, and the last three were those from the autoencoder. (**F**) Same as E for models with larger bottlenecks (64×4×4).

**Figure 4 entropy-23-01403-f004:**
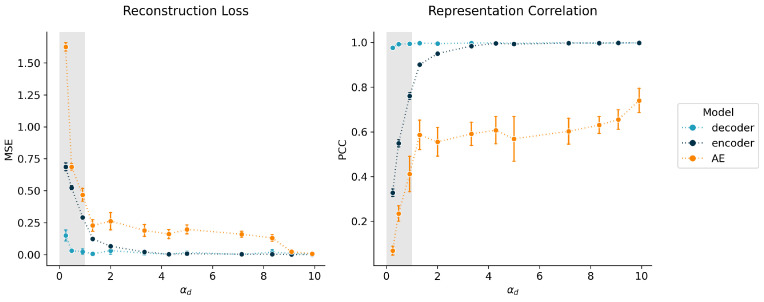
Performance metrics of decoder, encoder, and autoencoder for different decoder loads. Points and error bars present average losses and standard deviations over 5 replicates, respectively. Colors indicate the model type. Metrics were derived from the test data. **Left**: mean squared error (MSE) loss of the simulated data for different decoder loads. **Right**: Pearson correlation coefficients (PCCs) of true regulation dimension and model representation dimension for different decoder loads.

## Data Availability

Publicly available datasets were analyzed in this study. They include MNIST [13] and CIFAR10 [14]. Other data were randomly generated and can be reproduced using the description in Section 3.3 and the Appendix A available.

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
