# Peer review of "A Manifold Learning Perspective on Representation Learning: Learning Decoder and Representations without an Encoder"

_entropy, 2021, doi:10.3390/e23111403_

Round 1

Reviewer 1 Report

This is definitely one of the most clearly written papers which I encountered in 15 years of scientific career.

I have just two very minor comments.

1) The simulated data described in section 3.3 are probably clear to somebody who knows about "biological regulatory networks", which I do not master. It would be very helpful to illustrate in more details what these data look like, in order to give insights to a broader audience. I have a bit of trouble following this section.

2) It would be very interesting to read a short opinion / comment about the applicability of the results on problems involving time series and LSTM or transformer networks as building blocks. There are quite some activities going on in the field of using autoencoders to create representations of time signals for various purposes.

Author Response

We thank the reviewer for the nice introductory comment. Below we reply to the detailed comments

The simulated data described in section 3.3 are probably clear to somebody who knows about "biological regulatory networks", which I do not master. It would be very helpful to illustrate in more details what these data look like, in order to give insights to a broader audience. I have a bit of trouble following this section.

We have made a much more extensive description (at lines 320-325 in the old version).

It would be very interesting to read a short opinion / comment about the applicability of the results on problems involving time series and LSTM or transformer networks as building blocks. There are quite some activities going on in the field of using autoencoders to create representations of time signals for various purposes.

As LSTM- and Transformer-based autoencoders have successfully been applied to tasks such as time-series forecasting (Essien and Giannetti (2019) and Lim et al. (2020) as examples), we are optimistic in our assumption that these models can also be reduced to the decoder with at least the same performance as the full autoencoders. Depending on the data available and the complexity of the model, it may very well even lead to improvements in performance.

We added some text in the conclusion (line 377 in the orig. manuscript): Albeit only covering the task of reconstruction in this demonstration, we believe that this approach can easily be extended to other tasks in need of a useful representation. One such example is time series forecasting, based on building blocks such as LSTMs and Transformers.

Reviewer 2 Report

The paper presents a novel view on the relationship encoder/decoder architectures have with manifold learning and representation learning. The relevant concepts are presented in a clear and didactic way, easy to follow.

The relevant information, that the decoder can be trained by itself, without the explicit need for a decoder, doesn´t seem to me to be original in this work, but it is certainly better explained here than in any other source I was able to find. Overall, I think the work is very interesting and deserving of publication.

One issue is that all tests were performed on reconstruction tasks, which are the more natural ones in the context. Still, it would be interesting to understand if this approach (separate decoder training) also performs well in classification tasks or other types of tasks.

Some minor comments to be addressed by the authors:

Line 130: this approximation to the value of C_d in the case of MLPs seems a very rough approximation, in general. Some comments on the effects of this (possibly severe) overestimation could help here. Somehow, it seems there could be a relation here to some well-known measures of classifier complexity, but I understand that is outside the scope of the paper.

Lines 229 to 235: this explanation, although possibly correct, seems a bit handwaving to me and raises the question on whether there is something that deserves an additional analysis.  I still do not understand why the separate encoder training does not suceed, in the situation where the encoder is severely "under-specificed", which I assume happens here. It would be nice to have some additional information why training the encoder with the decoder fixed does not work. Maybe the optimization problem is too hard, of has local minima, among other possible explanations. 

Reviewer 3 Report

The subject needs to be consistent to the potentials, in the scientific field of manifold learning, as a quickly-growing subfield of machine learning and a clearer description on the internal structure of the autoencoders. Also, it needs better comparison with the state of the art.
